# Remote Sensing Monitoring of Ecological-Economic Impacts in the Belt and Road Initiatives Mining Project: A Case Study in Sino Iron and Taldybulak Levoberezhny

Yue Jiang [1], Wenpeng Lin [1,2,*], Mingquan Wu [3,4], Ke Liu [1,5], Xumiao Yu [1] and Jun Gao [1,2]

[1] School of Environmental and Geographical Sciences, Shanghai Normal University, Shanghai 200234, China; 1000496184@smail.shnu.edu.cn (Y.J.); 1000497640@smail.shnu.edu.cn (K.L.); 1000480063@smail.shnu.edu.cn (X.Y.); gaojun@shnu.edu.cn (J.G.)

[2] Yangtze River Delta Urban Wetland Ecosystem National Field Observation and Research Station, Shanghai 200234, China

[3] State Key Laboratory of Remote Sensing Science, Aerospace Information Research Institute, Chinese Academy of Sciences, Beijing 100094, China; wumq@aircas.ac.cn

[4] University of Chinese Academy of Sciences, Beijing 100049, China

[5] College of Life Sciences, Shanghai Normal University, Shanghai 200234, China

* Correspondence: linwenpeng@shnu.edu.cn

**Abstract:** Under the Belt and Road Initiatives, China's overseas cooperation in constructing mining projects has developed rapidly. The development and utilization of mining resources are essential requirements for socio-economic development. At the same time, the ecological impacts of the exploitation and utilization of mining resources have increasingly aroused the widespread concern of the international community. This paper uses Landsat images, high-resolution images, and nighttime light (NTL) data to remotely monitor Sino Iron in Australia and Taldybulak Levoberezhny in Kyrgyzstan in different development periods to provide a reference for the rational development of mineral resources and environmental management. The results show that the Chinese enterprises have achieved good results in the ecological protection of the mining area during the construction period. The development of the mine has caused minor damage to the surrounding environment and has not destroyed the local natural ecological pattern. The different NTL indices show an overall rising trend, indicating that the construction of mines has dramatically promoted the socio-economic development of countries along the Belt and Road in both time and space. Therefore, relevant departments should practice green development in overseas projects, establish a scientific mine governance system, and promote a win-win economic growth and environmental governance situation.

**Keywords:** the belt and road initiatives; surface mining; remote sensing monitoring; ecological environment; socio-economic

## 1. Introduction

In 2013, the Chinese government proposed the Silk Road Economic Belt joint construction and the 21st century Maritime Silk Road in Kazakhstan and Indonesia, respectively [1,2]. The Belt and Road Initiative (BRI) aims to strengthen mutually beneficial cooperation among countries along the route and promote stable regional development [3,4]. The Initiative has provided a new direction for the action of world politics and economy, received positive responses from all countries globally, especially the majority of developing countries, and has become the most promising international cooperation platform recognized by the international community [5,6]. Along with the BRI, countries along the route have become essential partners, trade markets, and investment objects in China's foreign trade, investment, and engineering construction. According to the "China Foreign Investment Cooperation Report 2020", China's investment in countries along the BRI is

diversified in content and industrial forms, with a series of cooperation in constructing and developing railroads, highways, ports, airports, industrial parks, energy, mines, and other fields. In 2019 alone, China signed nearly 7000 contracted overseas projects worth more than USD 150 billion [7]. In a report on the global economic impact of China's BRI, the Centre for Economics and Business Research (CEBR) states that the BRI will drive global GDP growth of more than USD 70,000 billion annually until 2040. Further, by 2040, nearly 56 countries will see their GDP grow by more than USD 10 billion due to the Initiative. The BRI has contributed significantly to the sustainable development of local societies and economies and the United Nations 2030 Sustainable Development Goals (SDGs) achievement worldwide [8].

Mineral resources are an essential material basis for social development, and almost all industrial sectors in modern society have some connection with the consumption of mineral resources [9]. Socio-economic development essentially depends on large amounts of mineral raw materials. Cooperation in mineral resources is an essential part of the Initiative. The countries along the BRI are rich in mineral resources and the world's leading supply base of mineral resources and raw materials [10]. Still, they often face inadequate infrastructure, underdeveloped industries, and low productivity, complementing China's more developed mineral mining equipment and technology. Mining cooperation can promote the development of countries in mineral resources exploration and help countries along the route improve the construction of roads, railroads, and other land transportation infrastructure and boost the growth of the local social economy [10,11]. However, the continuous exploitation of mineral resources and the deposition of mining waste will lead to various secondary environmental problems such as vegetation degradation, land occupation, ground subsidence, and biodiversity loss [12–14], which will pose challenges to the local ecological environment and the sustainable development of the social economy [15–18]. Relevant departments need to adopt more effective and reasonable monitoring tools and systems for project evaluation to better understand the impact of China's overseas projects on the ecological environment and socio-economy.

The traditional mining area survey method is a mainly manual survey, which requires a lot of human and material resources, has long duration, and a low efficiency. It cannot meet the needs of extensive area surveys and intuitive monitoring of dynamic changes in information [19–21]. On the other hand, the absence of data dramatically limits the development of targeted strategies to mitigate site-specific socio-environmental impacts [22,23]. Remote sensing technology has the characteristics of multi-temporal, large-scale synchronous monitoring and high-resolution information, which can provide solutions for collecting primary geographic data such as ecology, environment, resources, transportation, and topography in the BRI region. It has unique and irreplaceable advantages in ecological and environmental protection and socio-economic research in mining areas [24–26]. It can realize the monitoring of ecological and environmental information and inter-annual changes to a large extent and save costs, improve efficiency, provide timely and accurate information support to participating enterprises, governments, and the general public, and show the construction results.

Satellite-based remote sensing techniques are widely applied for evaluating and monitoring mining disturbance detection and environmental impact assessment [27,28]. The analysis of land cover change using Landsat and Moderate-resolution Imaging Spectroradiometer (MODIS) images by researchers can complete the characterization of mined land and surrounding landscape changes to monitor ecological environment changes [29–31]. Vegetation indices such as the Normalized Difference Vegetation Index (*NDVI*) and the Enhanced Vegetation Index (EVI) were used to assess the environmental environment in mining areas [32,33]. Machine learning algorithms such as support vector machines and random forests have been used to study land use classification in mining areas [34]. Scholars on this subject have carried out a great deal of research and practical work. Jhanwar et al. studied the changes in vegetation cover and mining area in the Bijolia mining area of Rajasthan while analyzing the impact of mining on the health of residents [35]. Yang

et al. monitored the vegetation disturbance and recovery status under a long time series of the Curragh coal mine in Australia [29]. Redondo-Vega et al. analyzed the changes in land use due to mining in the north-western mountains of Spain during the previous 50 years. The results show that the extraction of mineral resources through surface mining all causes damage to the previous topography and that these alterations are irreversible [36]. Ang et al. used Google Earth Engine and web mapping to study land cover changes and socio-environmental impacts of the Didipio gold and copper mining landscapes in the Philippines [37]. Xiao et al. used Google Earth Engine and the LandTrendr algorithm to map annual land disturbance and reclamation in a surface coal mining region of the Shengli Coalfield in Inner Mongolia from 2003 to 2019 [38].

The research on remote sensing monitoring of mining areas mainly focuses on ecological environment monitoring [39], such as vegetation change analysis, feature identification, extraction, etc. Still, there is less research on the socio-economic impact caused by the development and construction of mining areas. There is a lack of comprehensive research on mining areas' ecological environment and socio-economic implications. This is due to difficulty identifying and obtaining the indicators used to measure the socio-economic effects. However, with the development of night light remote sensing technology, it has been widely used in socio-economic and regional development studies, providing a more convenient and effective method for the economic monitoring of mining areas [40–43]. Yu et al. detected surface anthropogenic activities by tracking the nighttime light (NTL) data change from 1992 to 2013 [44]. Shi et al. used the NTL data to analyze the spatiotemporal patterns of electric power consumption in more than 60 countries and 20 major cities along the BRI from 1992 to 2013. The results reflected that electric power consumption during the 20 years of the BRI was mainly concentrated in developing countries and showed a northwest–southeast distribution [45].

In the past, domestic research on remote sensing monitoring applications of mining areas mainly focused on domestic mines, and the research objects were mostly coal mines [21,31,38]. There was less research on remote sensing monitoring overseas mines and other mining projects. The number of mining projects invested in by China overseas gradually increased after the proposed BRI. These collaborations are facing a series of problems while significantly promoting the social development of the countries concerned [46]. First, the relevant departments of China's overseas engineering projects' supervision are mainly based on regular statistical surveys, combined with questionnaires and other forms, lacking active monitoring technical means. The feedback information is easily subject to human interference and is not strong in real-time, strengthening the supervision effect. Secondly, at present, it mainly relies on questionnaire survey methods for statistics of relevant construction results [47], which has poor timeliness, primarily socio-economic data, and the presentation of results is mainly in the form of table data and other problems. At the same time, the World Bank and other western institutions and countries have widely adopted remote sensing and GIS technologies for presenting and evaluating the effects of an investment in projects, which can show the relevant results more intuitively and concretely. In addition, the impact of China's investment and construction along the BRI on the local environment is one of the focuses of international media attention [10,48,49]. Individual enterprises do not pay attention to protecting the local ecological environment, which affects the international image of China, thus leading to construction projects facing questions about environmental protection. Therefore, this study takes two major mining projects of the BRI as the research objects and uses multi-source remote sensing data to extract the land-use condition and vegetation coverage of the surrounding areas before and after the construction of the mine, monitor and analyze the ecological resources' occupation and loss, environmental restoration, and the impact of the structure of the mine on regional socio-economic development based on the NTL indices. This study aims to correctly understand and objectively evaluate the negative impact of mine development on the surrounding ecological environment to provide a reference for how to promote environmental protection in mining areas and sustainable development of the regional

economy. On the other hand, the research results are expected to provide a solid scientific basis and data support for implementing the BRI.

## 2. Materials

### 2.1. Study Area

This paper selects Sino Iron on the 21st century Maritime Silk Road and Taldybulak Levoberezhny on the Silk Road Economic Belt as the study areas to analyze the economic and ecological impacts of mining cooperation in the BRI (Figure 1). Sino Iron is located at Cape Preston, 100 km southwest of the town of Karratha in the Pilbara region of Western Australia [50]. The area has a dry climate and poor soil; it is one of the driest parts of the world and is dotted with laterite hills. The poor geological conditions and the hardness of the land made it difficult to work on construction. In addition, groundwater is scarce and of poor quality, resulting in higher construction costs for desalination facilities [51]. As an essential raw material for the iron and steel industry, iron ore has a high economic value and is an international commodity and strategic material. Sino Iron is the largest magnetite mining and operation project in Australia and one of the essential mines in China–Australia mining cooperation. The mine was developed in January 2007 by CITIC Pacific and Metallurgical Corporation of China (MCC). Since 2013, the mine project has supplied high-grade magnetite concentrates to CITIC Special Steel mills and other Chinese steel mills. With a mine life of more than 25 years, Sino Iron will enter operation with approximately 3000 direct and contractor employees, more than 95% of whom will be Australian residents. The project will provide mutual and extensive economic and social benefits to Australia and China for decades to come [51].

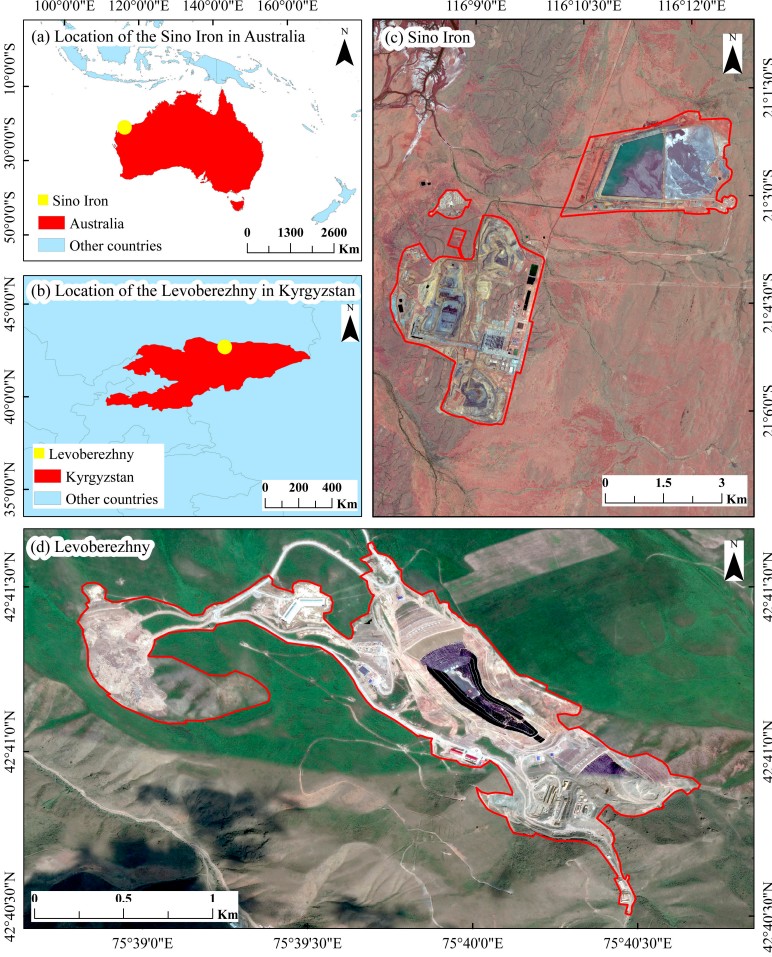

**Figure 1.** The location of Sino Iron and Taldybulak Levoberezhny.

Taldybulak Levoberezhny is located in the eastern part of the Northern Tien Shan in Kyrgyzstan [52], 120 km east of the capital city of Bishkek and 12 km south of the town of Orlovka. The north side of the mine is 26 km away from Kemin railway station, and there is a road connecting the mine area, which is more convenient for transportation. The mine area is 4–5 km long and 2 km wide. Taldybulak Levoberezhny is the second largest gold mine in Kyrgyzstan, one of the most significant mining investment projects of Chinese enterprises in Kyrgyzstan [53]. The mine was invested in by The Zijin Mining Group Co., Ltd. (Longyan, China) in 2011 and put into trial production in 2015, ending its 20 years of inactivity [52,54], which Kyrgyzstan's national leaders highly recognized. According to the plan, after the project is completed and reaches production, it can produce 3.7 tons of gold annually, create an output value of 150 million US dollars, solve the employment of 1000 local people, and contribute 24 million US dollars in tax revenue annually. It will significantly promote friendly relations and economic and cultural exchanges between the two countries [55].

*2.2. Data Sources*

This study uses long-term observations of multi-source (with different spatial resolutions and revisited intervals) remote sensing images, including high-resolution images in Google Earth, Landsat Thematic Mapper (TM)/Operational Land Imager (OLI), Defense Meteorological Satellite Program Operational Linescan System (DMSP-OLS) stable nighttime light, and Suomi National Polar-orbiting Partnership Visible Infrared Imaging Radiometer Suite (NPP-VIIRS) nighttime light for mining areas in recent years. The Landsat series of remote sensing imagery was used to extract and analyze the ecological resource types of the mine site and was obtained free of charge from the United States Geological Survey [56]. The spatial coverage of high-resolution imagery in Google Earth has expanded rapidly in recent years. Many high-resolution images (i.e., resolutions better than 1 m) have been captured and made available to track detailed changes in land cover [57]. The Google Earth images of 2010 (before the MCC put in construction) and 2020 (after the MCC put in the building) are selected for Sino Iron. The Google Earth images of 2010 (before investment by Zijin Mining) and 2016 (before investment by Zijin Mining) are chosen for Taldybulak Levoberezhny, which can extract more accurate land-use types within the mine area by manual visual interpretation. The NTL data are obtained from the 2000–2020 global 500 m resolution NPP-VIIRS-like nighttime light dataset produced by Chen et al. based on a deep learning model [58]. This dataset extends the length of time available for NTL data to extract changes in lighting in the area around the mine and thus to study economic impacts.

*2.3. Data Pre-Processing*

Landsat images need to be pre-processed before use, mainly for geometric correction, atmospheric correction [59], image cropping, and other processes to meet the accuracy requirements and for different time phases of the image cropping in the same size range as the study area. High-resolution images from Google Earth were visually interpreted to monitor the land-use change within the mine site. With the help of visual and other auxiliary information (including tone, texture, shapes, etc.) from high spatial resolution images, we have classified the land-use types within the mine site into classes such as open-pit, dump, road, contaminated mine site, and auxiliary production land. Figure 2 shows the visual interpretation results of different periods of Sino Iron and Taldybulak Levoberezhny. The NTL dataset is produced using an auto-encoder model-based cross-sensory calibration scheme and verified for accuracy to reflect detailed information about the inner city. This study requires cropping the NTL images and counting information such as the total brightness of the pixels and the total number of pixels with illumination in the study area. Figure 3 shows the NTL images of remote sensing monitoring areas of Sino Iron and Taldybulak Levoberezhny in different periods.

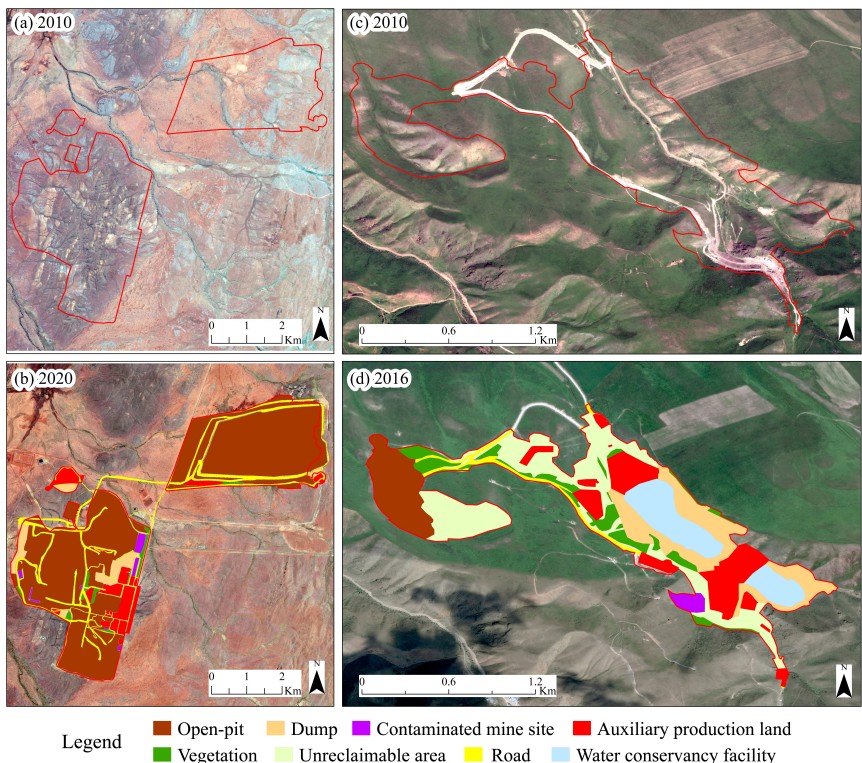

**Figure 2.** The visual interpretation results in different periods (**a**,**b**) for Sino Iron and (**c**,**d**) for Taldybulak Levoberezhny.

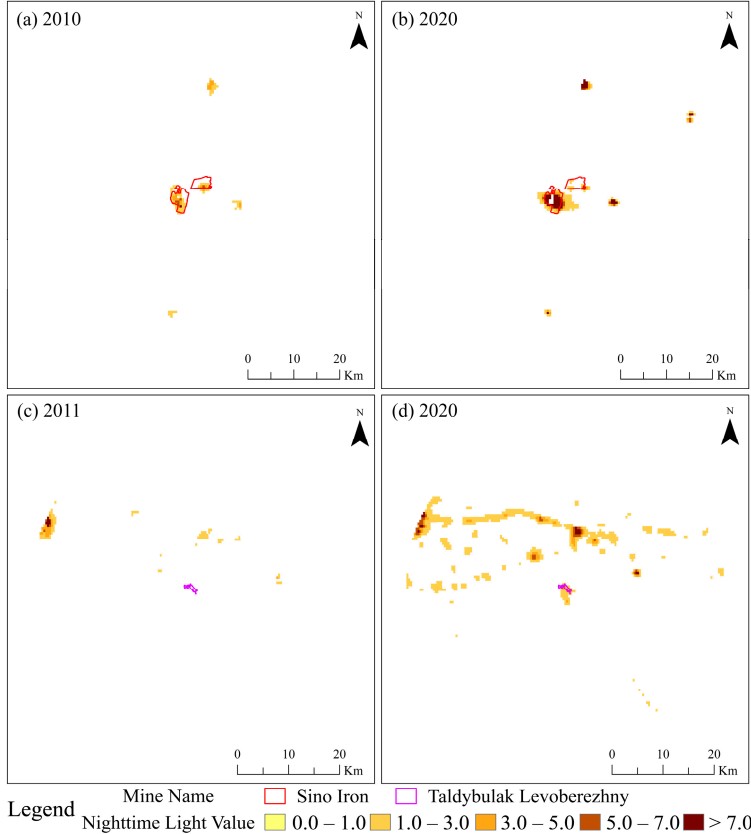

**Figure 3.** The nighttime light images of the monitoring area in different periods (**a**,**b**) for Sino Iron; (**c**,**d**) for Taldybulak Levoberezhny.

## 3. Methods

Based on the spatial and spectral properties of the satellite dataset, various image processing and interpretation techniques were applied to track the ecological and socio-economic impacts of mining activities throughout the entire time series. In the following sections, we summarize the methods used in this study.

### 3.1. Remote Sensing Monitoring of Ecological Environment

### 3.1.1. Vegetation Cover Extraction

The vegetation cover is the percentage of the vertical projection of vegetation on the ground to the region's total area, which can effectively reflect the characteristics of surface vegetation distribution and monitor the changes in the ecological environment. Mining usually causes changes in surface vegetation, including deforestation and, in reclaimed areas, afforestation, which the dynamic changes of vegetation cover can monitor [44,60]. In this study, the regional *NDVI* was calculated by Equation (1), and then the vegetation cover information was extracted by conversion using Equation (2).

$$NDVI = \frac{N_{NIR} - N_R}{N_{NIR} + N_R} \tag{1}$$

$$VFC = \frac{NDVI - NDVI_{soil}}{NDVI_{veg} - NDVI_{soil}} \tag{2}$$

where $N_{NIR}$ and $N_R$ are the surface reflection values in the infrared and red bands, respectively [61]. *NDVI* values always range between $-1$ and $+1$, where negative values reflect the absence of vegetation, the higher index values being associated with greater green leaf area and biomass [62]. $NDVI_{soil}$ is the *NDVI* value of the area covered by bare soil; $NDVI_{veg}$ is the *NDVI* value of the room surrounded by vegetation. In this paper, for $NDVI_{soil}$ and $NDVI_{veg}$ values, the maximum and minimum values of the given confidence interval are taken according to the grayscale distribution of the whole *NDVI* image in the study area and the actual condition of the vegetation cover in the mining area. Intercepting the *NDVI* frequency distribution table at a 5% confidence level, the upper and lower thresholds of *NDVI* were taken as $NDVI_{soil}$ with a frequency of 5% and $NDVI_{veg}$ with a frequency of 95%.

Based on the characteristics of the adopted Landsat series satellite multispectral image data, this study extracted the vegetation cover information of the 500 m, 1 km, 5 km, and 10 km buffer zones around the mine area in two different periods. Referring to the Standards for Classification and Gradation of Soil Erosion (SL190-2007), the Technical Rules of Land Use Survey (1984), and previous research results [9,63] to set reasonable thresholds (see Table 1), the growth status of vegetation was classified.

**Table 1.** The vegetation cover class classification table.

| Level | Vegetation Cover | Vegetation Cover Class |
|:---:|:---:|:---:|
| I | [0–0.2] | Low vegetation cover |
| II | [0.2–0.4] | Sub-Low vegetation cover |
| III | [0.4–0.6] | Moderate vegetation cover |
| IV | [0.6–0.8] | Sub-High vegetation cover |
| V | [0.8–1.0] | High vegetation cover |

### 3.1.2. Identification of Characteristic Surface Types in Mining Areas

Identifying characteristic surface types in mining areas is the most extensive research in remote sensing of the mining ecological environment. The primary research is to identify and categorize various surface types and then monitor the changes of different kinds in the regional scope, an essential indicator for monitoring the ecological environment. Based on the visual interpretation results of Google Earth high-resolution images, this study analyzes the impact of the construction of Sino Iron and Taldybulak Levoberezhny on the surrounding

ecological environment by comparing the area of each surface type and its percentage change in two different periods before and after the Chinese enterprises started construction.

### 3.1.3. Landsat-Based Ecological Resource Changes Detection

The Landsat-based mining ecological resources change detection includes three main steps: pre-processing the Landsat time series, image classification, and change analysis based on the results [44]. Step 1 is described in detail in Section 2.3. Step 2: we established remote sensing monitoring areas around Sino Iron and Taldybulak Levoberezhny. The supervised classification algorithm was applied to complete the ecological resource change analysis of Landsat images collected through training sample collection, accuracy verification, maximum likelihood method classification, and research results. Ecological resources include vegetation, bare land, water, and mining land. This study used the confusion matrix approach to verify the accuracy of classification results with Kappa coefficients and overall classification accuracy. The results are shown in Table 2, and the overall classification accuracy of both mines was above 80%. Step 3: the ecological resource distribution maps at different periods and the statistical results of each resource type's area were used to study the impacts on the ecological environment around the mining area.

**Table 2.** Accuracy evaluation.

| Mine Name | Year | Overall Classification Accuracy | Kappa Coefficient |
|---|---|---|---|
| Sino Iron | 2010 | 95.8452% | 0.9133 |
| | 2020 | 95.5215% | 0.9049 |
| Taldybulak | 2011 | 81.2932% | 0.6042 |
| Levoberezhny | 2020 | 89.8362% | 0.7644 |

### *3.2. Remote Sensing Monitoring of Economic Impact*

The NTL index is one of the most critical indicators in remote sensing science to measure socio-economic development, which has a significant correlation with economic growth and can also monitor the speed of urban construction and the driving force of urban expansion. This study took the mining area as the monitoring center and the surrounding 30 km buffer zone area as the study area. Then different NTL indices within the study area were calculated to analyze the impact of the mine construction on the local socio-economy. The following formula calculates the total NTL index and the average NTL index [64,65].

$$L_{total} = \sum_{i=1}^{n} DN_i \tag{3}$$

$$L_{avg} = \frac{L_{total}}{\sum_{i=1}^{n} i} \tag{4}$$

where $L_{total}$ is the total NTL index, $L_{avg}$ is the average NTL index, $DN_i$ denotes the *DN* value of the *i*-th pixel, and *n* represents the total number of pixels in the study area.

The NTL area index refers to the total area of all bright pixels as a percentage of the entire study area, reflecting the extension of the lights. In addition, previous studies show that a *DN* value greater than seven is generally a more economically developed area. The calculation formulas are as follows.

$$S = \frac{S_N}{S_A} \times 100\% \tag{5}$$

$$Area_{gt7} = c \times \sum_{i=1}^{j} x_j \tag{6}$$

where *S* is the NTL area index, $S_N$ is the total area of all bright pixels, and $S_A$ is the total area of the study area. $Area_{gt7}$ indicates the total area of the pixels with a *DN* value greater than seven, *c* is a constant representing the area of a pixel, *j* is the number of pixels with a *DN* value greater than seven, and $x_j$ is the *j*-th pixel with the *DN* value greater than seven.

## 4. Results

### *4.1. Influence of Ecological Environment in Mining Area*

#### 4.1.1. Time-Series Changes in Vegetation Cover

This paper studies the influence of mine construction on the surrounding vegetation. The spatial distribution of vegetation in the remote sensing monitoring areas of the two mines during different periods is shown in Figure 4. Both Sino Iron and Taldybulak Levoberezhny selected two phases of remote sensing images before and after the Chinese enterprises started building to calculate the area percentage of different vegetation cover classes in the buffer zone of different distances around the mine area. The statistical results are shown in Figure 5.

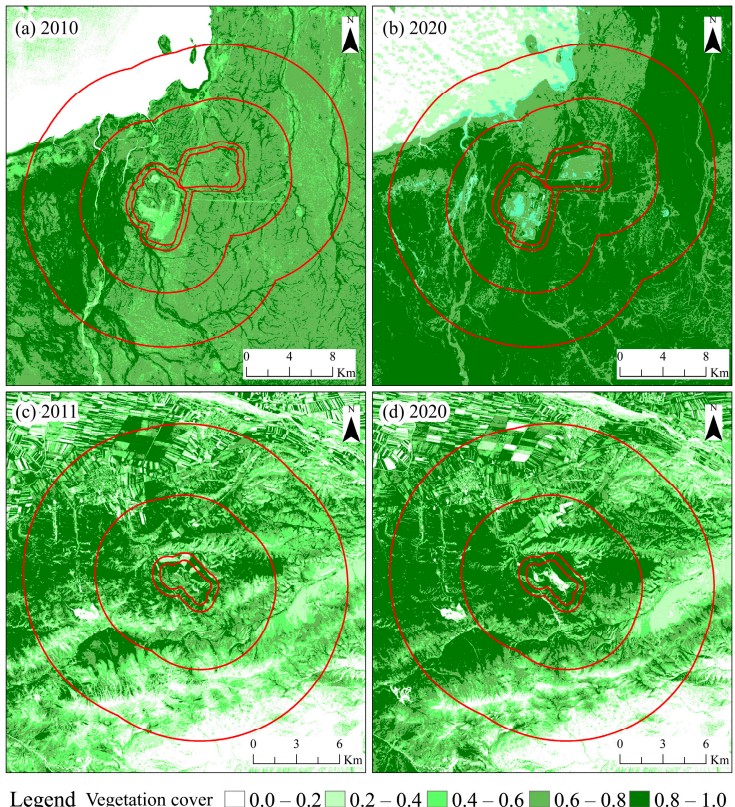

**Figure 4.** Distribution of vegetation cover in different periods in mining areas (**a**,**b**) for Sino Iron and (**c**,**d**) for Taldybulak Levoberezhny.

In 2010, all the buffer areas in Sino Iron were dominated by Class IV vegetation cover. In 2020, except for the 500 m buffer zone, which was still dominated by Class IV vegetation cover, the other three buffer zones in Sino Iron were dominated by Class V vegetation cover. Overall, the area of vegetation covering Class IV and Class V in 2020 was increasing in each buffer zone, and the increase was more prominent, with the sum of the two accounting for more than 70%. In contrast, the area of Class III vegetation cover in each buffer zone in 2020 has decreased compared with that in 2010. The area share of vegetation covers Class I and Class II within each buffer zone fluctuated within a small range due to the exploitation of mineral resources within the mine area and its adjacent areas, thus resulting in its low vegetation cover. As can be seen from Figure 5, the area of high vegetation cover in the buffer zones of 500 m, 1 km, 5 km, and 10 km in 2020 for Taldybulak Levoberezhny increased compared with 2011 by 14.03%, 18.17%, 17.72%, and 9.78% respectively. In contrast, sub-low vegetation cover, moderate vegetation cover, and sub-high vegetation cover areas have all decreased. In addition, except for a slight decrease in the area of low vegetation cover area within the 10 km buffer, the area increased within the other three buffer zones, especially within the 500

m buffer zone, which increased by 10.36%. As the distance of the mine buffer zone increases, the overall vegetation area increases, and the vegetation coverage increases.

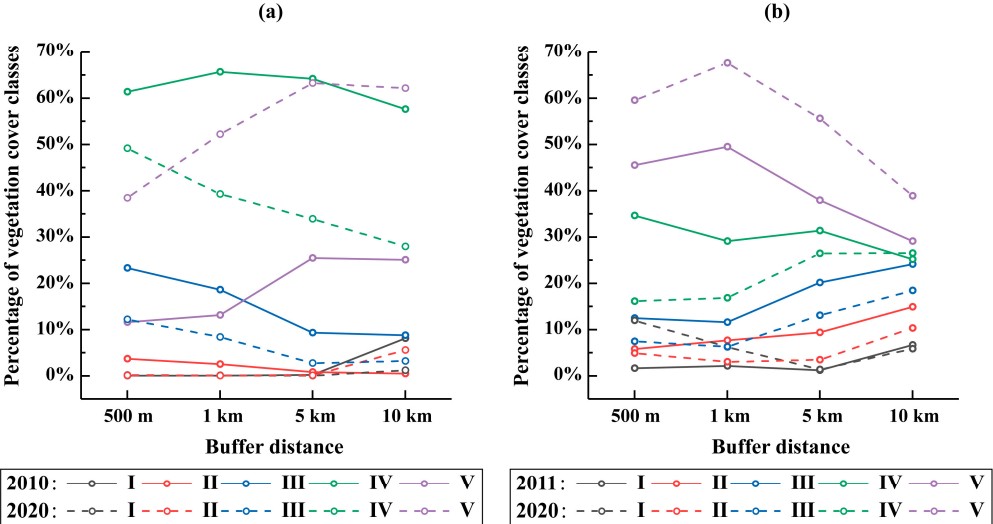

**Figure 5.** Changes in the area percentage of different vegetation cover classes in different buffer zones (**a**) for Sino Iron and (**b**) for Taldybulak Levoberezhny.

### 4.1.2. Time-Series Changes of Characteristic Surface Types in the Mining Area

The visual interpretation results (Figure 2) and statistical data (Tables 3 and 4) are used to analyze the changes in the characteristic surface types of the mining area. Since the structure of Sino Iron has not been carried out since 2010, there were no mining areas and other land-use types in this monitoring area. After carrying out the construction of Sino Iron in the period from 2010 to 2020, the scope of each land-use type showed significant changes. With the increase in mining efforts, the open-pit area increased by 15.09 km$^2$ in 2020 compared to 2010, accounting for 73.349%. The dump area was developed to 0.839 km$^2$ compared to before construction, accounting for 4.039% of the entire mine. The vegetation size in the mine site was 0.30 km$^2$, accounting for 1.45%. The extent of land for auxiliary production facilities, roads, and contaminated sites in the mine was 1.49, 2.64, and 0.23 km$^2$, respectively, and their percentages were 7.239%, 12.84%, and 1.11%, respectively. In terms of Taldybulak Levoberezhny, it is clear that the mining area was in a state of pending development before Zijin Mining invested in its construction. With Zijin Mining's investment in Taldybulak Levoberezhny, the mining started to work smoothly. The open-pit area was 0.29 km$^2$, accounting for 10.46% of the mine, indicating that the mine is rich in mineral resources and has a large proportion of gold mining. The land-use types such as dump, water conservancy facilities, and land for auxiliary production facilities within the mine area accounted for 42.06% of the study area. In addition, the size of the contaminated mine site was 0.04 km$^2$, occupying 1.43% of the mine. The subsequent mining work should protect the ecological environment, and the land reclamation work still needs more effort.

**Table 3.** Changes in the characteristic surface types of Sino Iron from 2010 to 2020.

| Year | Land Use | Open-Pit | Dump | Vegetation | Auxiliary Production Land | Road | Contaminated Mine Site |
|---|---|---|---|---|---|---|---|
| 2010 | Area (km$^2$) | 0 | 0 | 0 | 0 | 0 | 0 |
| | Percentage (%) | 0 | 0 | 0 | 0 | 0 | 0 |
| 2020 | Area (km$^2$) | 15.09 | 0.83 | 0.30 | 1.49 | 2.64 | 0.23 |
| | Percentage (%) | 73.34 | 4.03 | 1.45 | 7.23 | 12.84 | 1.11 |
| 2010–2020 | Area (km$^2$) | 15.09 | 0.83 | 0.30 | 1.49 | 2.64 | 0.23 |
| | Percentage (%) | 73.34 | 4.03 | 1.45 | 7.23 | 12.84 | 1.11 |

**Table 4.** Changes in the characteristic surface types of Taldybulak Levoberezhny from 2010 to 2016.

| Year | Land Use | Open-Pit | Dump | Vegetation | Auxiliary Production Land | Contaminated Mine Site | Water Conservancy Facilities | Unreclaimable Area | Road |
|---|---|---|---|---|---|---|---|---|---|
| 2010 | Area (km²) | 0 | 0 | 0 | 0 | 0 | 0 | 0 | 0 |
| | Percentage (%) | 0 | 0 | 0 | 0 | 0 | 0 | 0 | 0 |
| 2016 | Area (km²) | 0.29 | 0.41 | 0.23 | 0.42 | 0.04 | 0.34 | 0.73 | 0.33 |
| | Percentage (%) | 10.46 | 14.71 | 8.10 | 15.10 | 1.43 | 12.25 | 26.06 | 11.89 |
| 2010–2016 | Area (km²) | 0.29 | 0.41 | 0.23 | 0.42 | 0.04 | 0.34 | 0.73 | 0.33 |
| | Percentage (%) | 10.46 | 14.71 | 8.10 | 15.10 | 1.43 | 12.25 | 26.06 | 11.89 |

### 4.1.3. Ecological Resource Occupation and Restoration

Figure 6 shows the distribution of ecological resources in the remote sensing monitoring region of Sino Iron and Taldybulak Levoberezhny. Table 5 shows the results of monitoring the environmental resource changes of Sino Iron. The results show that in 2010, the area around the mine was dominated by bare land, with an area of 305.11 km², accounting for 86.47%. The distribution was relatively concentrated and appeared in patches. The extent of vegetation was 32.64 km², accounting for 9.25% of the total area of the monitoring area. The water area was 13.29 km², accounting for 3.77%. By 2020, the ecological resources were still mainly bare land, with an area of 239.21 km², accounting for 67.79%, a decrease of 65.90 km² or 18.68% compared to 2010. The extent of vegetation reached 61.66 km², accounting for 17.48%, an increase of 29.02 km² or 8.23% compared with 2010. The mining land area in 2020 was 38.10 km², accounting for 10.80% of the total area of the remote sensing monitoring area. Table 6 shows the ecological resource monitoring results of Taldybulak Levoberezhny, indicating that the area around the mine was dominated by vegetation and bare land. In 2011, when Zijin Mining did not invest in construction, the vegetation area was 422.55 km², accounting for 84.83%, and was concentrated in the mine's western and northern regions. The bare land area was 63.83 km², accounting for 12.81%, and focused on the mine's southeast site. In addition, residential areas were concentrated in the northern part of Taldybulak Levoberezhny, a total of 11.74 km², accounting for 2.36% of the entire monitoring area. With construction from Zijin Mining, the ecological resources around Taldybulak Levoberezhny have also changed. The mining land has increased by 1.39 km² in 10 years, accounting for 0.29% of the whole monitoring area, indicating that the mining work is progressing more smoothly, and the scale of mining is gradually expanding. The residential area increased to 14.22 km², 0.49% higher than the share in 2011.

**Table 5.** Ecological resources in the monitoring area of Sino Iron.

| Year | Type | Vegetation | Bare Ground | Water | Mining Land | Total |
|---|---|---|---|---|---|---|
| 2010 | Area (km²) | 32.64 | 305.11 | 13.29 | 0 | 351.04 |
| | Percentage (%) | 9.25 | 86.47 | 3.77 | 0 | 99.49 |
| 2020 | Area (km²) | 61.66 | 239.21 | 12.06 | 38.10 | 351.04 |
| | Percentage (%) | 17.48 | 67.79 | 3.42 | 10.80 | 99.49 |

**Table 6.** Ecological resources in the monitoring area of Taldybulak Levoberezhny.

| Year | Type | Vegetation | Bare Ground | Mining Land | Residential Area | Total |
|---|---|---|---|---|---|---|
| 2011 | Area (km²) | 422.55 | 63.83 | 0 | 11.74 | 498.12 |
| | Percentage (%) | 84.83 | 12.81 | 0.00 | 2.36 | 100.00 |
| 2020 | Area (km²) | 423.02 | 59.49 | 1.39 | 14.22 | 498.12 |
| | Percentage (%) | 84.92 | 11.94 | 0.29 | 2.85 | 100.00 |

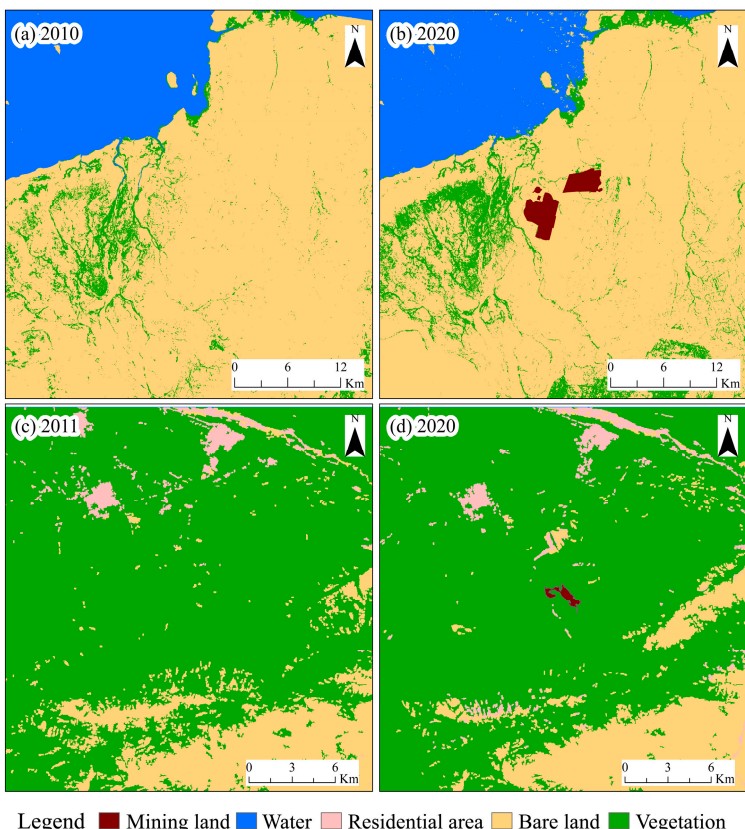

**Figure 6.** Distribution of ecological resources in the monitoring area (**a**,**b**) for Sino Iron and (**c**,**d**) for Taldybulak Levoberezhny.

### 4.2. Influence of Economic Condition in Mining Area

Figure 7 shows the variation of different NTL indices within the remote sensing monitoring area of Sino Iron and Taldybulak Levoberezhny from 2001 to 2020. The changes in the NTL indices in the monitoring area of Sino Iron can be divided into four specific phases. From 2001 to 2007, we did not detect any NTL information in the region. The total NTL index, the average NTL index, the NTL area index, and the total area of the pixels with a DN value greater than seven for the above four indicators were maintained at 0. Combined with the land-use classification results, the area was undeveloped wasteland during this period. From 2008 to 2010, the total NTL index increased from 0 to 279.8905, the average NTL index rose from 0 to 0.0179, and the growth of these two indicators was relatively slow. The NTL area index also showed an increasing trend, but the difference was that its growth rate was somewhat faster than the first two. For the total area of the pixels with a DN value greater than seven, this indicator did not change. From 2011 to 2013, all the above four indicators showed a rapid growth trend, and the total area of the DN value greater than seven changed most significantly, from 0.25 to 14.25 km$^2$. From 2014 to 2020, these four indicators showed a fluctuating growth trend, but the growth rate slowed down and stabilized. In addition, taking 2007 as the time point, it can be divided into two periods. Before the investment of Chinese Enterprises, all NTL indices were 0; after the acquisition, the total NTL index, the average NTL index, the NTL area index, and the total area of the pixels with the DN value greater than seven were 1077.9468, 0.0758, 0.9327% and 9.3077 km$^2$, respectively. It can be seen that the construction of Sino Iron by CITIC Pacific and MCC has created a pulling effect on the development of the local economy.

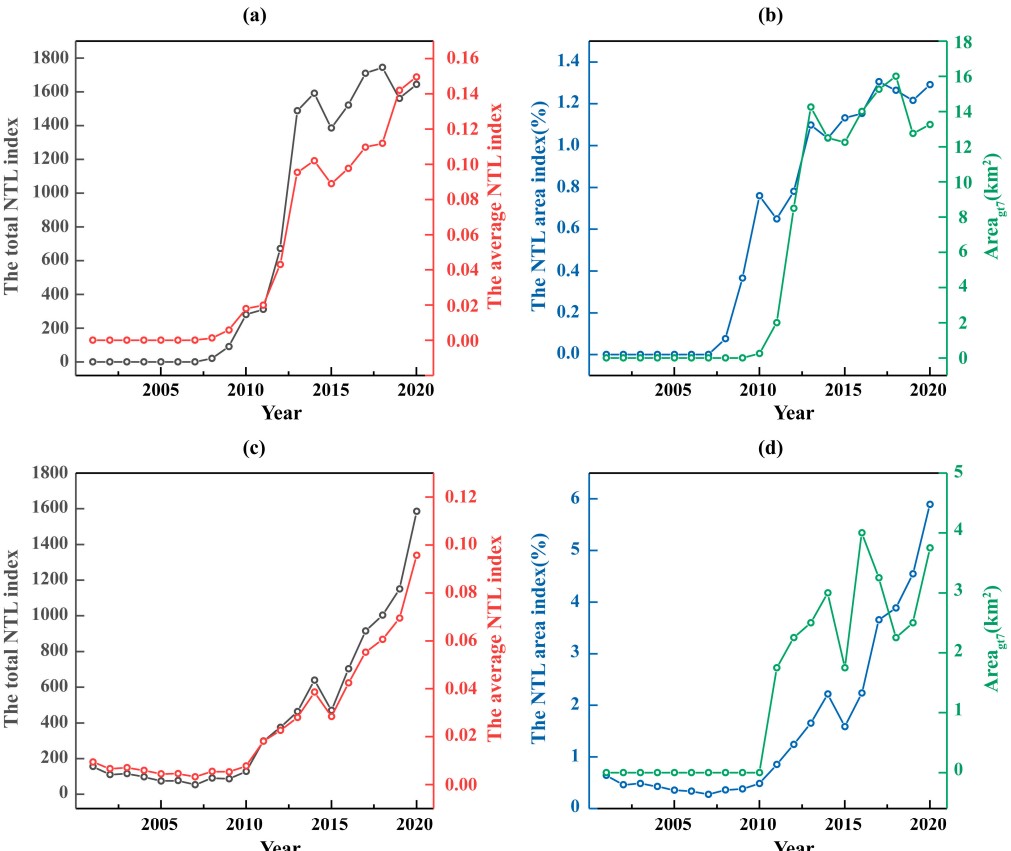

**Figure 7.** Trends of different NTL indices in the monitoring area from 2001 to 2020 (**a,b**) for Sino Iron and (**c,d**) for Taldybulak Levoberezhny.

The results of remote sensing monitoring of NTL index changes in the monitoring area of Taldybulak Levoberezhny can be divided into three stages. From 2001 to 2010, the total NTL index, the average NTL index, and the NTL area index were maintained at low levels. The indicator of the total area of the pixels with a DN value greater than seven was always 0, indicating that the site did not belong to a more economically developed region. From 2011 to 2014, all four indicators showed a growth trend, with average annual growth rates of 37.9454%, 37.9368%, 53.2051%, and 23.8095%, respectively. From 2015 to 2020, differently from the previous period, the growth fluctuated, but the overall trend was of continuous increase. In addition, taking 2011 as the time point can be divided into two periods. Before Zijin Mining's investment, the mean values of the total NTL index, the average NTL index, the NTL area index, and the area with a DN value greater than seven are 98.5790%, 0.0059%, 0.4199%, and 0 km$^2$, respectively. While in the period after Zijin Mining's construction, the mean values of the four indices were 760.0204, 0.04589, 2.7768%, and 2.7 km$^2$, respectively. The above results showed that the development level of the area around Taldybulak Levoberezhny had improved.

## 5. Discussion

Since the proposed BRI, China's foreign engineering projects have developed rapidly. However, the lack of primary data and insufficient understanding of socio-economic conditions have led to failed foreign investment projects in China. The environmental impacts caused by engineering projects have also drawn increasing attention from the international community [10]. However, the research related to the construction of overseas cooperation projects of the BRI is still relatively weak. The relevant studies are mainly strategic and general. Some scholars have published preliminary studies on the spatial distribution pattern of energy [66], trade structure [67], and mineral resources development potential [68]. The available studies mainly focus on energy strategy and ecological assessment, while

fewer serve the specific construction of infrastructure projects, challenging to meet the relevant needs of offshore engineering construction. This study uses multi-source remote sensing images to achieve dynamic monitoring of overseas mining projects' ecological and economic impacts. Compared with related studies, this study can make full use of remote sensing data, the required data are easier to obtain, and the evaluation indexes and methods are efficient, which enables rapid monitoring of the distribution, development trend, ecological-economic impacts, and construction progress of the overseas projects along the BRI and has achieved good application practice. In the future, we will further improve the evaluation indexes and build a systematic index system to quickly, efficiently, and accurately supervise overseas projects and provide data support for safeguarding China's strategic material reserves.

As the project continues to advance, the intensity of mining has increased, increasing the area of open-pit and contaminated sites, indicating some damage to the ecological environment around the mine. However, the distribution and changes of vegetation cover in the monitoring area show that the overall vegetation condition shows a trend of improvement, and the ecological environment is developing in a good direction. Comprehensive analysis shows that Chinese enterprises have adopted green construction methods in the construction of mines [69], paying more attention to protecting the local ecological environment and reducing the impact of engineering construction, which is conducive to the region's sustainable development [38]. In addition, with the background of global co-governance and the continuous implementation of the concept of sustainable development [70], it is believed that Chinese enterprises will carry out the Green Mine concept in the construction of mines in the future and contribute to the sustainable development of the regional economy. This paper has certain shortcomings in studying the impact on the ecological environment. First, in identifying the characteristic surface types in mining areas, there is an inevitable subjectivity in the classification due to the criteria of the classification system and the connotation definition of each name, plus the differences in the characteristics of the mining areas themselves. Second, due to the lack of a large-scale high-resolution image sample library of typical surface types in mining scenarios [71], this study still relies on manual sample selection and self-construction of a minimal training sample dataset. This severely limits the accuracy of surface type recognition in mining areas and the ability of the algorithm to generalize different images in different study areas, thus limiting the large-scale application of cutting-edge advanced intelligence algorithms [72]. Third, due to data accessibility and classification accuracy limitations, this study only selected indicators of vegetation cover, characteristic surface types of mining areas, and surrounding ecological resources for ecological environment monitoring, which is not comprehensive enough. It is worth exploring how to select more integrated and comprehensive indicators for scientific and accurate tracking of mine projects in subsequent studies [73].

In terms of socio-economic impact, the area and distribution of roads and construction land (including auxiliary production land, water conservancy facilities, etc.) are influenced by mining intensity. The streets are densely distributed throughout the mine area, extending along the open-pit area. At the same time, the auxiliary production land has been expanded to ensure the smooth development of the production work, indicating that the development of the mine area can promote the development of infrastructure [74]. On the other hand, with the continuous development of the mine project, the local NTL indices have increased significantly in terms of spatial extension and intensity, indicating the speed and scale of economic growth around the mine area increased. This is because the source of NTL comes from the lighting of human activities at night. The increase in the NTL indices in the mining area indicates that the development of the mining project drives the rise in the area of related industrial construction land, improving the employment opportunities for the people and increasing the market opportunities. The above analysis shows that under the BRI, China's overseas engineering projects have brought considerable benefits to the economies of the regions where the mines are located and have played a role in promoting the economic development of the countries along the route [10,75]. Only a

single indicator of NTL is used to assess the economic benefits of mines in our study, and its representativeness is subject to further discussion. In addition, the NTL dataset utilizes a deep learning model to achieve cross-sensor correction, which involves different satellites and sensor types with different resolutions and still has some errors [58], which will impact the study results. The launch of high-resolution NTL remote sensing satellites, such as Jilin-1 and Luojia-1, can provide NTL data with sub-meter resolution on a global scale [76–78]. We expect the development analysis of the Belt and Road and other regions based on NTL remote sensing will be more accurate.

Future remote sensing monitoring of mining areas can use more ecological parameters for comprehensive assessment, such as chlorophyll content of vegetation, the heavy metal content of vegetation, and atmospheric pollutants (e.g., PM2.5 and PM10), to explore the impact of mining on the ecological environment from the perspective of pollution in-depth, and to provide an effective supplement to the existing research ideas on environment changes in mining areas [79]. In addition, we can try to use global grid products such as GDP distribution and population distribution to improve the efficiency and accuracy of comprehensive development level assessment in the follow-up study [80,81].

## 6. Conclusions

In 2013, the BRI was proposed to highlight the geographical advantages of the countries along the route and bring them closer together to form a Community with a Shared Future for Humankind. The development model states that clear waters and green mountains are as good as mountains of gold and silver has put forward higher and more scientific requirements for coordinating and matching the regional economic structure and the ecological environment. This study investigates the environmental impacts of mining activities in representative countries along the BRI, assesses the economic benefits of mines, and discusses the gaps in the existing research and challenges associated with follow-up studies.

Countries along the BRI are rich in mineral resources, which are crucial for economic development, but at the same time, ecological disturbances caused by mining are inevitable. The pressure from the environment and energy demand means that mining must change its inherent ways. In implementing overseas projects, it is necessary to implement the concept of Green Mines and adopt green construction methods to minimize the impact on the surrounding ecology and consider incorporating the ecological restoration of mines into the BRI.

**Author Contributions:** Conceptualization, W.L. and M.W.; Data curation, W.L.; Formal analysis, Y.J., K.L. and X.Y.; Funding acquisition, W.L.; Investigation, W.L.; Methodology, Y.J., K.L. and X.Y.; Project administration, W.L.; Resources, M.W.; Software, Y.J., W.L., K.L. and X.Y.; Supervision, W.L.; Validation, Y.J., W.L., M.W., K.L., X.Y. and J.G.; Visualization, Y.J., K.L. and X.Y.; Writing—original draft, Y.J.; Writing—review and editing, Y.J., W.L., K.L., X.Y. and J.G. All authors have read and agreed to the published version of the manuscript.

**Funding:** This work was supported by the National Natural Science Foundation of China (Nos. 41730642 and 41571047).

**Data Availability Statement:** The data used to support the findings of this study are available from the corresponding author upon request.

**Acknowledgments:** We thank all the participants involved in the project for their contribution to our research data.

**Conflicts of Interest:** The authors declare no conflict of interest.

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
