# Peer review of "Remote Sensing Monitoring of Ecological-Economic Impacts in the Belt and Road Initiatives Mining Project: A Case Study in Sino Iron and Taldybulak Levoberezhny"

_remotesensing, doi:10.3390/rs14143308_

Round 1
Reviewer 1 Report
The paper has been corrected according to my previous comments and in my opinion it can be published in the current form.
Author Response
We appreciate all the suggestive and valuable comments. Those comments are very helpful to the thorough revision and improvement of our paper and also give us essential guidance for further research.
Reviewer 2 Report
Dear authors, thank ypou very much for improving your manuscript. Kind regards,
Author Response

(The authors gave the same response as above.)

Reviewer 3 Report
Even if the manuscript presents some improvements and aspects that could be topical, the background of the research is insufficient. The topic is not a current one, it has been researched and debated in numerous other articles, practically the authors go on an old path and do not bring anything new.
Author Response
We thank the reviewer for the constructive comments. Point-to-point responses to each comment are as follows.
[Comment 1]: Even if the manuscript presents some improvements and aspects that could be topical, the background of the research is insufficient.
Response: We appreciate the positive and valuable comments. Following your suggestions, we have carefully revised our manuscript. In the introduction section, we have added the research background accordingly and cited relevant references. In the past, domestic research on remote sensing monitoring applications of mining areas mainly focused on domestic mines, and the research objects were mostly coal mines. There was less research on remote sensing monitoring overseas mines and other mining projects. The number of mining projects invested in China overseas gradually increased after the proposed Belt and Road initiative. These collaborations are facing a series of problems while significantly promoting the social development of the countries concerned. For example, the relevant departments in China do not have enough supervisory capacity, the statistics on the Belt and Road construction results are not in place, and the construction of projects faces questions about ecological and environmental protection. Therefore, this study takes two major Belt and Road mining projects as the research objects and provides new ideas, technologies, methods, and means to solve the above challenges by dynamically monitoring overseas mining projects' ecological-economic impacts.
[Comment 2]: The topic is not a current one, it has been researched and debated in numerous other articles, practically the authors go on an old path and do not bring anything new.
Response: Thank you for your comments. Although there has been researched on the use of remote sensing technology to investigate mining resources, further studies are needed on using satellite remote sensing and other active monitoring means to supervise the Belt and Road overseas engineering projects. Since the Belt and Road initiative, China's foreign engineering projects have developed rapidly. In 2020, China's total imports and exports to countries along the Belt and Road reached 9.37 trillion. China's enterprises along the 61 countries signed 5611 new contracts for foreign contracting projects, with a newly signed contract value of $141.46 billion. Therefore, the rapid development of China's overseas engineering construction has put higher requirements for the research on project planning and design, sustainable building and development, risk assessment and prevention, economic benefits assessment, development situation analysis and operation supervision, etc. With the further promotion of the Belt and Road Initiative, the efficient management of construction projects and environmental protection is becoming more critical, making this study necessary. Compared with related studies, this study can make full use of remote sensing data, the required data are easier to obtain, and the evaluation indexes and methods are efficient, which enables rapid monitoring of the distribution, development trend, ecological-economic impacts, and construction progress of the overseas projects along the Belt and Road, and has achieved good application practice. In the future, we will further improve the evaluation indexes and build a systematic index system to quickly, efficiently, and accurately supervise overseas projects and provide data support for safeguarding China's strategic material reserves.

Round 2
Reviewer 3 Report
The authors have improved the manuscript with the needed modifications.
This manuscript is a resubmission of an earlier submission. The following is a list of the peer review reports and author responses from that submission.
Round 1
Reviewer 1 Report
The paper “Remote Sensing Monitoring of Socio-Economic and Ecological Resources in the Belt and Road Initiatives Mining Project: A case study in the Sino Iron and the Levoberezhny” presents an interesting research which fits to the scope of the journal, however, it should be revised before publishing. Below I present some comments which should be taken into consideration.
- I suggest to highlight an international context of the research. Introduction could refer to more (recent) studies from researchers from other countries who focus in their studies on monitoring based on remote sensing data. This in especially important considering that there is no separate section “Literature review”, and current version of the Introduction is very brief.
- Quality of some figures should be improved.
- Data included in table could be easier to analyse by presenting them on charts. I suggest to include them especially for the most crucial data that influence conclusions and findings of the research.
- The paper is missing on of the basic elements of a scientific paper – “Discussion” section (one of element in an IMRaD structure). It should include a comparative discussion of the obtained results with other studies. It could be compared for example with other approaches (including automated or semi- automated) of land use / land cover analysis. Considering a time scope used in the submitted research (2010-2020 / 2011-2020) and results presenting change over time, it could be used also for preparation of forecasts of future trends in land use / land cover change. See for instance: A framework for path-dependent industrial land transition analysis using vector data, European Planning Studies, Volume 27, 2019 - Issue 7
I encourage the Authors to correct the paper, as in my opinion it presents an interesting study and might constitute a valuable paper after improvements mentioned above.
Reviewer 2 Report
Dear authors, I am enclosing a few brief suggestions to improve the quality of your manuscript:
1.- Please standardize the format of the references according to the requirements of Remote Sensing and MDPI.
2.- I recommend that you reduce the length of the manuscript title. A title should be concise and to the point, giving the reader just enough information.
3.- In the introduction, I advise you not to mention the name of any politician, you can mention the government or the agencies you consider appropriate, but not individuals. On the other hand, please, at the end of the introduction, write more clearly the main objective of the research carried out, and its possible applications.
4.- When describing the study area, please complete the description with geological, climatic and edaphic data of the area, with the corresponding bibliography.
5.-
The main recommendation for your manuscript is the following: you should include a correct discussion of your results. You already have such a discussion mixed with your own data, but you have not used references to contrast your information. Please, in the results only put the data obtained, and generate a new discussion section, where you contrast your results with others already published and expose the limitations and applications of your study. All this using the appropriate bibliographic references.
6.- Conclusions should be more precise. Shorten this section.
Kind regards,
Reviewer 3 Report
The paper needs a revision of some sentences because are too long. E.g. from the line, 90 to 102 it is too long and not clear. therefore, therefore, there is a need to clarify better the aim of the study.
The method structure has to be clarified, E.g I didn't understand how you have used and why formula 2 and formula 4.
Moreover, it is not clear the link between NDVI analysis and SNL analysis
But I think, that after modification, the manuscript can be published